# Purchasing under threat: Changes in shopping patterns during the COVID-19 pandemic

**Sebastian Schmidt** [1]*, **Christoph Benke** [1], **Christiane A. Pané-Farré**[1,2]

**1** Department of Psychology, Clinical Psychology, Experimental Psychopathology, and Psychotherapy, Philipps University Marburg, Marburg, Germany, **2** Center for Mind, Brain and Behavior (CMBB), Philipps University Marburg and Justus Liebig University Giessen, Gießen, Germany

* sebastian.schmidt.psychologie@uni-marburg.de

**Data Availability Statement:** All files (data set, R code) are available from the data_UMR repository under the following URL: https://data.uni-marburg.de/handle/dataumr/110.

## Abstract

The spreading of COVID-19 has led to panic buying all over the world. In this study, we applied an animal model framework to elucidate changes in human purchasing behavior under COVID-19 pandemic conditions. Purchasing behavior and potential predictors were assessed in an online questionnaire format ($N$ = 813). Multiple regression analyses were used to evaluate the role of individually *Perceived Threat of COVID-19*, anxiety related personality traits (trait-anxiety, intolerance of uncertainty) and the role of media exposure in predicting quantity and frequency of purchasing behavior. High levels of *Perceived Threat of COVID-19* were associated significantly with a reported reduction in purchasing frequency ($b$ = -.24, $p$ < .001) and an increase in the quantity of products bought per purchase ($b$ = .22, $p$ < .001). These results are comparable to observed changes in foraging behavior in rodents under threat conditions. Higher levels of intolerance of uncertainty ($b$ = .19, $p$ < .001) and high extend of media exposure ($b$ = .27, $p$ < .001) were positively associated with *Perceived Threat of COVID-19* and an increase in purchasing quantity. This study contributes to our understanding of aberrated human purchasing behavior and aims to link findings from animal research to human behavior beyond experimental investigations.

## Introduction

The spreading of the coronavirus disease (COVID-19) has led to worldwide stockpiling of food and hygiene products which caused temporally shortages [1]. In early March 2020, when the number of daily COVID-19 infections reached its peak in Germany [2], the German Federal Statistical Office recorded an enormous increase in sales of goods of sanitary and daily needs [3]: e.g., early in March 2020, a 150% increase for pasta, 153% for soap, and 751% for disinfectants. Similar changes in shopping behavior were recorded in the USA [4] and the UK [5]. At the same time, studies indicated an increase in fear and worries related to the virus [6, 7].

The modulation of foraging behavior by threat has extensively been studied in the animal model [8]. In the natural environment, animals need to ensure a sufficient calorie intake while

**Funding:** The authors received no specific funding for this work.

**Competing interests:** The authors have declared that no competing interests exist.

trying to avoid predatory attack. To parallel the natural habitat, animal studies use a safe nest area that must be left to obtain food. To evaluate threat related changes in foraging, the animals are confronted with a threat stimulus in the foraging area, such as the smell of a predator [9] or an electric shock [8]. In response to such threat encounter animals show an increase in risk assessment behaviors, e.g., attentive head-scanning [10], an inhibition of appetitive behavior [11], an increased latency in the procurement of food pellets [12] as well as a reduction in number of meals accompanied by an increase of the size of portions to maintain caloric intake [8].

A recent study investigated factors influencing stockpiling during the COVID-19 pandemic. Increased COVID-related worry (e.g., "I will become very ill."; "I will not have access to food.") was associated with stockpiling of more products indicating that negative affect like worries and anxiety influence shopping behavior [13].

In parallel to a predatory attack which constitutes a threat during natural foraging, the possibility of an infection with COVID-19 constitutes a threat in a human purchasing situation under pandemic conditions. In line with the described animal and human findings, we hypothesized that individually perceived threat resulting from possible COVID-19 infection will predict changes in human purchasing behavior under the current pandemic. Human purchasing is not only limited to food items. Increased selling rates were also reported for hygiene products such as disinfectant and toilet paper. Therefore, it seemed reasonable to consider purchasing of these necessities as a part of human foraging. Based on findings from animal research [8], we expected that perceived *Threat of COVID-19* will lead to (1) a reduction in purchasing frequency and (2) an increase in purchasing quantity per purchase.

Additionally, we were interested in the influence of other factors known to influence feelings of anxiety that thus might be associated with threat perception of COVID-19 and changes in purchasing behavior. It has been demonstrated that psychological vulnerability factors such as trait-anxiety (i.e., the tendency to experience anxiety and perceive situations as threatening) and intolerance of uncertainty (i.e., the tendency of an individual to experience possible negative future events as unacceptable and threatening) increase the risk to fearfully respond to potentially negative or uncertain stimuli, events or situations such as those arising during the current pandemic. Both psychological factors have been linked with occurrence of anxiety-related disorders [14–16]. Initial evidence from the current COVID-19 pandemic revealed that trait-anxiety and intolerance of uncertainty are associated with higher levels of threat perception and fear of the coronavirus [17, 18]. Another relevant factor that has been discussed to increase fear and threat perception of COVID-19 via transmission of threat information is the level of exposure to media. Studies from current COVID-19 pandemic higlight the role of increased media exposure on elevated anxiety and stress responses as well as increased fear of COVID-19 under the COVID-19 pandemic [17, 19]. In the present study, we tested whether *Perceived Threat of COVID-19* explains changes in purchasing behavior beyond these factors.

## Purpose of the present study and hypotheses

Understanding the causes for changes in consumers purchasing behavior under the COVID-19 pandemic is of high relevance for governments and policymakers, e.g., to avoid panic buying which in turn may cause shortage of important goods. As stated in a perspective article by Van Bavel and colleagues [20] there are several research topics relevant to the COVID-19 pandemic which have to be addressed by social and behavioral sciences. Fear is a central emotional response during a pandemic which shapes information processing (e.g., risk perception) and behavior (e.g., shopping behavior). Based on a theoretical framework derived from rodent foraging behavior under threat, we examined the role of perceived threat originating from the

present COVID-19 pandemic situation in predicting changes in purchasing patterns of groceries and hygiene products in an online questionnaire study. Our main hypotheses are that higher levels of *Perceived Threat of COVID-19* would be (1) associated with a reduction of purchasing frequency and (2) an increase in purchasing quantity per purchase. We also expected a positive correlation between *Perceived Threat of COVID-19* and an increase in purchasing quantity for individual products. In face of the known relevance of trait-anxiety and intolerance of uncertainty as risk factors for anxiety disorders and depression, we hypothesized that these constructs would be positively related to *Perceived Threat of COVID-19*. Additionally, we expected individuals with high vulnerability to develop anxiety disorders (high trait-anxiety, high intolerance of uncertainty) to show a decrease in shopping frequency and an increase in purchasing quantity per purchase. Besides we hypothesized that being part of a risk group for a severe course of an infection with COVID-19 or having regular contact with a high-risk person would be associated with higher levels of *Perceived Threat of COVID-19* and changes in shopping behavior as described above. A high extend of media exposure was also hypothesized to be positively associated with *Perceived Threat of COVID-19* and changes in purchasing patterns (increased purchasing quantity while reducing shopping frequency).

## Methods

The study was conducted from April 23[rd] to May 18[th], 2020. In this time window the total amount of confirmed COVID-19 cases in Germany had reached 175.896. The implementation of public health measures by the German federal states started in March 2020 (e.g., prohibition to meet with others in public places, closure on non-essential shops, or closure of kindergartens or daycare institutions [21] while risk communication increased in the media, e.g., daily report of case numbers or information that infection with COVID-19 may cause a life-threatening disease and recommendations on how to avoid infection [19]. In effect, for March 2020 massive increases in sales figures were reported [3]. We asked participants to retrospectively rate their purchasing behavior for this month. The online questionnaire was realized using SoSci Survey [22] and was published on soscisurvey.de (see supplementary information for a German (S1 Appendix) and an English version (S2 Appendix) of the questionnaire).

## Participants

In total 1074 individuals completed the online questionnaire and gave an answer to every question. Participants who did not finish the questionnaire were excluded. Data analysis was further limited to those participants for whom buying groceries constituted an actual risk of COVID-19 infection at time of assessment, i.e., we excluded participants who had already gone through a COVID-19 infection (*n* = 3), or did not actually visit any stores during the assessment period due to either being in quarantine (*n* = 30) or exclusively shopping online (*n* = 49). To achieve a valid assessment of purchasing behavior changes from pre-pandemic to pandemic, we also excluded participants who did not make their own purchases (because, e.g., the partner did) before (*n* = 65) and during the pandemic (*n* = 150) leaving a final sample of 813 respondents (78% female). Participants were aged between 18 and 79 (*M* = 42.42, *SD* = 15.00) (see Table 1 for descriptive statistics). The survey was advertised via the central e-mail system of Philipps University Marburg and on social media platforms. In order to motivate as many people as possible to participate in the online study, the raffle of three food delivery vouchers worth € 39.99 each was announced. The study was approved by the ethics committee of the Department of Psychology at the Phillips University of Marburg. Participants were informed that participation is voluntary and can be cancelled at any time without giving

**Table 1. Descriptive statistics.**

| Variable | % | Mean | SD |
|---|---|---|---|
| Sex (male) | 22 | - | - |
| Age | - | 42.42 | 15 |
| Educational Level [1–5] | - | 4.08 | 0.91 |
| Household Size | - | 2.40 | 1.56 |
| High Risk Person (Yes) | 41 | - | - |
| High Risk Loved (Yes) | 50 | - | - |
| Social Desirability Bias [7–28] | - | 19.86 | 2.33 |
| Perceived Threat of COVID-19 [1–7] | - | 4.16 | 1.36 |
| Intolerance of Uncertainty [12–60] | - | 32.5 | 9.92 |
| Trait Anxiety [20–80] | - | 40.31 | 11.13 |
| Risk Perception [0–100] | - | 25.68 | 25.57 |
| Media Exposure [1–4] | - | 3.2 | 0.72 |

$N$ = 813. Possible range of scores is given in parentheses. Coding for educational level: 1 = "no degree", 2 = "primary education", 3 = "secondary school diploma", 4 = "high school graduation", 5 = "university degree".

reasons, and that data will be stored anonymously. Written informed consent was obtained on the first page of the online questionnaire.

## Measures

**Predictors.** *Perceived Threat of COVID-19. Perceived Threat of COVID-19* was measured using six semantic differential seven-point rating scales. The six items were introduced with "*The novel coronavirus is for me. . ..*" following two oppositely poled adjectives, ("concerning" vs. "not concerning", "frightening" vs. "not frightening", "something I am thinking about all the time" vs. "something I am not thinking about all the time", "something I feel helpless about" vs. "something I can actively do something about", "burdensome" vs. "not burdensome", "close" vs. "far away"). These items were taken from the COSMO Snapshot Monitoring study conducted by the University of Erfurt [7]. The internal consistency of this scale was good (Cronbach´s $\alpha$ = .86). A principle component analysis indicated a one-dimensional construct, so we used the mean score of all six items as an indicator of perceived threat.

*Intolerance of Uncertainty Scale.* We used the 12-item short version of the *Intolerance of Uncertainty Scale* which maps the tendency of an individual to experience possible negative future events as unacceptable and threatening (e.g., "Unforeseen events upset me greatly.") and is associated with worry, state-anxiety and related to anxiety pathologies [23]. The reported internal consistency of the short version is Cronbach´s $\alpha$ = .91. The internal consistency in this sample was good (Cronbach´s $\alpha$ = .87). A German validation study reported similar results (Cronbach´s $\alpha$ = .90) and reported intolerance of uncertainty to be predictive for worrying [24].

*State Trait Anxiety Inventory.* We used the trait portion of the *State Trait Anxiety Inventory* (A-Trait, e.g., "I worry too much over something that really doesn´t matter.") which consists of 20 items. The internal consistency in this sample reached an excellent value of Cronbach´s $\alpha$ = .94. The reported Cronbach´s $\alpha$ for the A-Trait lies between .86 - .95 [25], for the German version Cronbach´s $\alpha$ = .90 [26].

*Risk Perception.* Participants assessed the likelihood of being infected with COVID-19 while shopping on a continuous scale ranging from 0% ("very unlikely") to 100% ("very likely").

*Extend of media exposure.* We asked participants to indicate how often they gather information about the COVID-19 pandemic on a four-point Likert scale (1 = "never", 2 = "less than once a day", 3 = "once a day", 4 = "several times a day").

*Risk Group.* Based on a standardized description (*"There is an increased risk of a severe course of COVID-19 disease for persons aged 50 years or older, smokers, persons with existing heart or lung diseases, chronic liver disease, diabetes mellitus, cancer or a weakened immune system."*) participants indicated if they (in person) belong to a risk group for a severe course of COVID-19 or if they have regular contact to a person (e.g., household member) belonging to such a risk group (coding: 0 = "no", 1 = "yes").

*Social Desirability Bias.* The *Scale for Detecting Test Manipulation through Faking Good and Social Desirability Bias* consists of seven five-level Likert items [27]. We used the individual scores to control for socially desirable reporting biases.

*Demographic Variables.* Participants reported their age in years, sex (coding: 0 = "female", 1 = "male"), in which federal state they live, their highest level of education (1 = "no degree", 2 = "primary education", 3 = "secondary school diploma", 4 = "high school graduation", 5 = "university degree") and their household size (number of persons living in a household).

**Outcome measures.** *Purchasing Behavior.* Participants indicated the change in purchasing frequency and change in purchasing quantity for the month March 2020 relative to January 2020. We used January 2020 as a reference because at that point the German government did not consider the coronavirus to be a risk for Germany [28], no infection control measures were implemented yet [29] and no changes in purchasing behavior were observed compared to the usual level [3]. Participants were able to indicate the full range of *change in purchasing frequency* on a seven-point rating scale: *Compared to January 2020, before the outbreak of the Corona pandemic in Germany, how often did you go shopping in March 2020*? (options: -3 = "much less frequently", -2 = "less frequently", -1 = "little less frequently", 0 = "just as often", 1 = "little more often", 2 = "more often", 3 = "much more often"). In correspondence, *change in purchasing quantity* was assessed using the following item: *Compared to January 2020, before the outbreak of the Corona pandemic in Germany, how much (quantity) did you buy per purchase in March 2020*? (options: -3 = "much less", -2 = "less", -1 = "a little less", 0 = "just as much", 1 = "a little more", 2 = "more", 3 = "much more").

*Purchasing Quantity for individual products.* For a more differentiated analysis we asked respondents to rate the purchasing quantity for individual products for March 2020 relative to January 2020. The following products were rated: toilet paper, soap, disinfectants, canned food, noodles/rice and fresh products (e.g., cheese, meat). There was the additional option to choose "do not usually buy this product".

## Data analysis

*Purchasing Frequency* and *Purchasing Quantity* were analyzed in separate multiple regressions controlled for gender, age, education, household size, and social desirability bias. In a next step, we entered all COVID-19 related variables (being part of a risk group, extend of media exposure to inform about COVID-19, risk perception of getting an infection, *Perceived Threat of COVID-19*) and anxiety related personality traits (intolerance of uncertainty, trait-anxiety) as a predictor of interest and examined its specific effect above the baseline model. In a final set of analyses, we entered all significant variables in one model and compared their effects. The same model was used to analyze the change in purchasing quantity for individual products.

Since *Perceived Threat of COVID-19* was our main predictor of interest, we conducted an additional multiple regression analysis with the same baseline model as explained above and

included the additional factors (e.g., sex, age, intolerance of uncertainty) to examine their specific predictive value for *Perceived Threat of COVID-19*. For the ease of interpretation all continuous variables were z-standardized before entered into the model. We checked for multicollinearity using the variance inflation factor (VIF). All VIFs were smaller than two and thus considered unproblematic [30]. Since the dependent variables (purchasing frequency and purchasing quantity) were not normally distributed, we decided to additionally report confidence intervals (95% CI) based on bootstrapping [31] to bypass the assumptions for multiple linear regression. 2000 samples were generated to obtain an empirical distribution (using the boot.ci-function from the R package "boot"). Note that these results were highly comparable to the results of the parametric test. Additionally, we report non-parametric analyses (e.g., ordinal logistic regression) for the main findings as supplementary information (see S3 Appendix), again showing highly comparable results. All analyses were conducted with R [32].

## Results

### Change in purchasing frequency

The distribution of participants' rating of change in purchasing frequency (see Fig 1) shows that 32.1% of study participants indicated that they went shopping for groceries as often in March as they did in January. 57.3% of the participants indicated that they went shopping less often and 10.6% indicated that they went shopping for groceries more often in March as compared to January. Overall, a one-sample *t*-test revealed a significant decrease in purchasing frequency from January to March ($M = -0.86$, $SD = 1.35$), $t(812) = -18.274$, $p < .001$, $d = 0.64$.

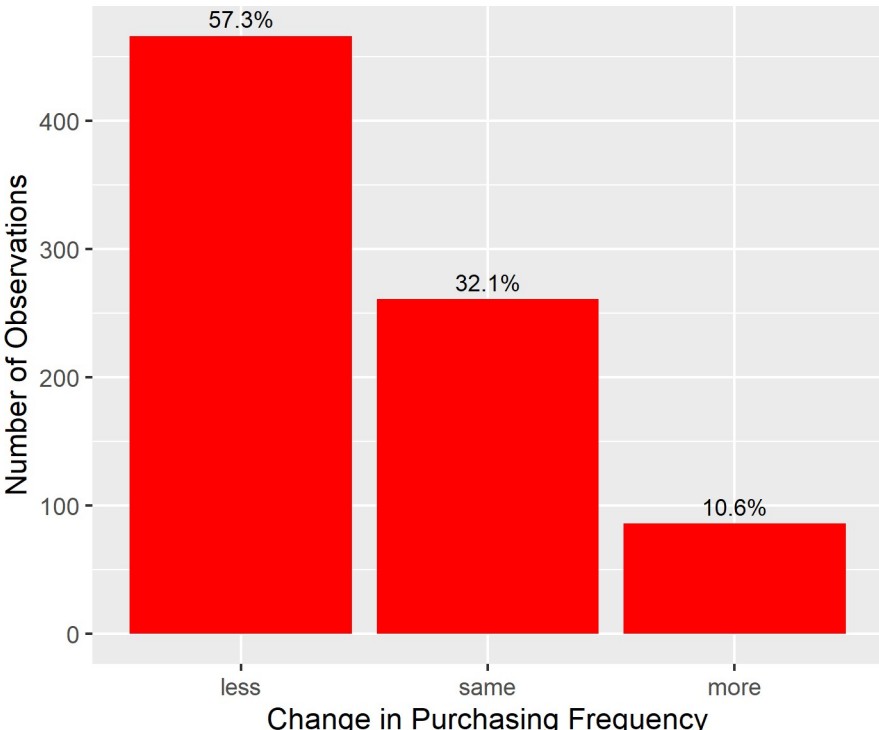

**Fig 1. Distribution of subjective change in purchasing frequency in March compared to January 2020.** $N = 813$. Note that categories "less" and "more" each comprise three gradations of the original scale (see section *Outcome Measures*).

Regression models were calculated for the full range scale (see S3 Table) and, for clarity of hypothesis testing regarding a *decrease* in foraging frequency, excluding those 10.6% of participants who report an increase in purchasing frequency (see text below). Bivariate correlations between all variables are presented in S1 and S2 Tables (full range scale) in the supporting information. In the baseline model (see Table 2) sex and educational level were the only significant predictors for purchasing frequency. Female sex was associated with a decrease in purchasing frequency in March 2020 compared to January 2020, $b = 0.32$, $t(672) = 3.46$, $p = .001$, 95% CI [.13, .49]. Higher education was associated with a reduction of purchasing frequency, $b = -.13$, $t(672) = 3.49$, $p = .001$, 95% CI [-.21, -.06]. Adding *Perceived Threat of COVID-19* to the model revealed that higher subjective threat was associated with a decrease in purchasing frequency, $b = -.30$, $t(671) = 8.21$, $p < .001$, 95% CI [-.39, -.23]. Intolerance of uncertainty and trait-anxiety revealed suppression effects, $b = -.08$, $t(671) = 2.12$, $p = .035$, 95% CI [-.16, -.01] respectively $b = -.10$, $t(671) = 2.57$, $p = .010$, 95% CI [-.18, -.03] (see S1 Table for correlations). The perception for being at high risk for infection with COVID-19 during shopping was associated with a decrease in purchasing frequency, $b = -.19$, $t(671) = 5.03$, $p < .001$, 95% CI [-.26, -.10]. Adding media exposure significantly improved the model, $b = -.18$, $t(671) = 4.62$, $p < .001$, 95% CI [-.25, -.10]. Belonging to a risk group was not a significant predictor of change in purchasing frequency ($b = -.17$, $t(671) = 1.92$, $p = .056$, 95% CI [-.35, .01]) nor was having regular contact with a risk person ($b = -.11$, $t(671) = 1.46$, $p = .145$, 95% CI [-.26, .05]).

Finally, to check whether *Perceived Threat of COVID-19*, risk perception and media exposure explained specific variance above and beyond the baseline model, all three predictors were entered in one block after the baseline model (see Table 3). The analysis revealed that *Perceived Threat of COVID-19* ($b = -.24$, $t(667) = 5.60$, $p < .001$, 95% CI [-.33, -.15]), risk perception ($b = -.10$, $t(667) = 2.63$, $p = .006$, 95% CI [-.18, -.02]) and media exposure ($b = -.11$, $t(667) = 2.78$, $p = .009$, 95% CI [-.19, -.03]) added incremental variance to the baseline model. The overall model explained 12.9% of the variance in change in purchasing frequency, $F(10, 667) = 11.07$, $p < .001$.

**Table 2. Prediction of change in purchasing behavior (frequency/quantity).**

| Predictors | Change in Purchasing Frequency | | | Change in Purchasing Quantity | | |
|---|---|---|---|---|---|---|
| | *b* | 95% CI$_{boot}$ | *p-value* | *b* | 95% CI$_{boot}$ | *p-value* |
| Sex | **.32** | **.13 – .49** | **.001** | **-.19** | -.37 – .02 | **.040** |
| Age | -.05 | -.12 – .03 | .237 | **-.09** | **-.17 – -.02** | **.018** |
| Educational Level | **-.13** | **-.21 – -.06** | **.001** | **.11** | **.03 – .19** | **.003** |
| Household Size | -.01 | -.09 – .07 | .730 | .02 | -.05 – .12 | .526 |
| Social Desirability Bias | .03 | -.05 – .10 | .477 | -.04 | -.12 – .04 | .278 |
| **Added Predictor** | | | | | | |
| Perceived Threat of COVID-19 | **-.30** | **-.39 – -.23** | **< .001** | **.29** | **.22 – .37** | **< .001** |
| Intolerance of Uncertainty | **-.08**[*] | **-.16 – -.01** | **.035** | **.11** | **.02 – .18** | **.007** |
| Trait-Anxiety | **-.10**[*] | **-.18 – -.03** | **.010** | **.10** | **.01 – .18** | **.016** |
| Risk Group (Self) | -.17 | -.35 – .01 | .056 | .03 | -.15 – .20 | .700 |
| Risk Group (Loved) | -.11 | -.26 – .05 | .145 | -.01 | -.16 – .13 | .943 |
| Media Exposure | **-.18** | **-.25 – -.10** | **< .001** | **.19** | **.12 – .26** | **< .001** |
| Risk Perception | **-.19** | **-.26 – -.11** | **< .001** | **.19** | **.11 – .27** | **< .001** |

Multiple linear regression performed on the restricted sample ($N = 678$). Table reveals the specific effect for a single predictor beyond the baseline model. Significant regression weights ($p < .05$) of the multiple regression analysis are printed in bold. All continuous variables were included as z-standardized variables. Dichotomous Variables: Coding for sex: female = 0, male = 1; coding for being part of a risk group for a severe COVID-19 disease course: no = 0, yes = 1. *Perceived Threat of COVID-19* was the best predictor for change in purchasing frequency ($R^2adj. = .12$) and change in purchasing quantity ($R^2adj. = .11$). [*]Suppressor Effect: Variable did not reveal a significant bivariate correlation with the criteria.

**Table 3. Overall model: Prediction of change in purchasing behavior (frequency/quantity).**

| Predictors | Change in Purchasing Frequency | | | Change in Purchasing Quantity | | |
|---|---|---|---|---|---|---|
| | b | 95% $CI_{boot}$ | p-value | b | 95% $CI_{boot}$ | p-value |
| Sex | **.21** | **.05 – .39** | **.019** | -.09 | -.25 – .10 | .332 |
| Age | -.04 | -.12 – .03 | .262 | **-.10** | **-.17 – -.02** | **.014** |
| Educational Level | **-.12** | **-.19 – -.05** | **.001** | **.10** | **.02 – .16** | **.008** |
| Household Size | -.01 | -.08 – .07 | .833 | .02 | -.04 – .11 | .609 |
| Social Desirability Bias | .00 | -.07 – .07 | .964 | -.02 | -.09 – .06 | .612 |
| Perceived Threat of COVID-19 | **-.24** | **-.33 – -.15** | **< .001** | **.22** | **.13 – .30** | **< .001** |
| Intolerance of Uncertainty | .05 | -.05 – .14 | .308 | -.01 | -.10 – .09 | .877 |
| Trait-Anxiety | -.04 | -.13 – .05 | .401 | .02 | -.08 – .11 | .744 |
| Media Exposure | **-.11** | **-.19 – -.03** | **.006** | **.12** | **.05 – .20** | **.002** |
| Risk Perception | **-.10** | **-.18 – -.02** | **.009** | **.11** | .03 – .20 | **.004** |
| $R^2$ / $R^2$ adjusted | .142 / .129 | | | .136 / .123 | | |

Multiple linear regression on the restricted sample ($N$ = 678). In this analysis all predictors adding significant variance beyond the baseline model were entered in one step. Significant regression weights ($p < .05$) of the multiple regression analysis are printed in bold. All continuous variables were included as z-standardized variables. Coding for sex: female = 0, male = 1.

## Change in purchasing quantity

The distribution of participants' rating of change in purchasing quantity (see Fig 2) shows that 45.5% of the participants indicated that they bought the same number of products per

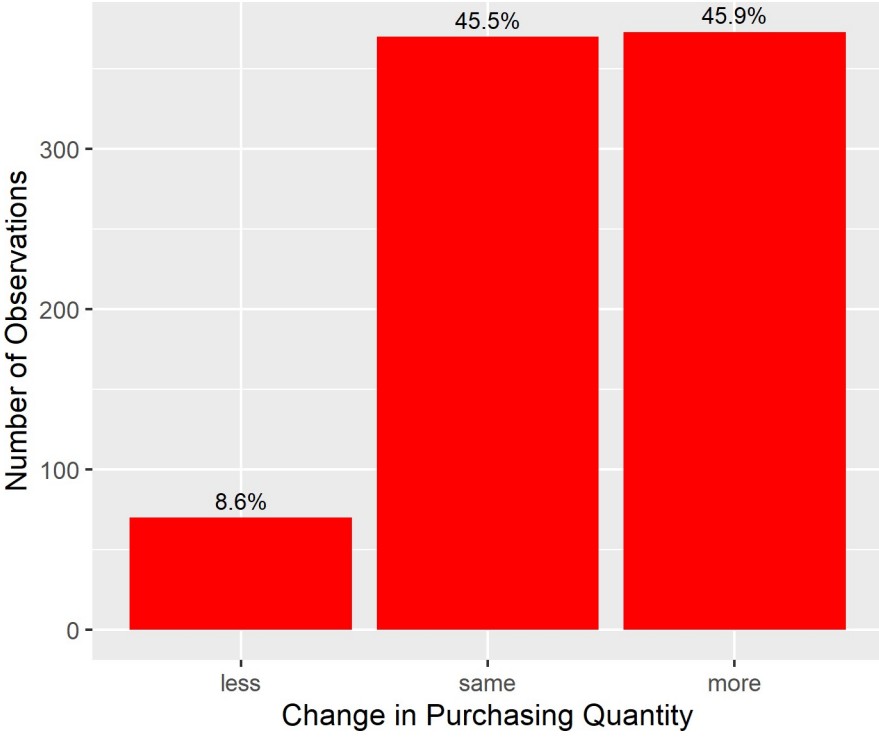

**Fig 2. Distribution of subjective change in purchasing quantity in March compared to January 2020.** $N$ = 813. Note that categories "less" and "more" each comprise three gradations of the original scale (see section *Outcome Measures*).

purchase in March as in January 2020. 8.6% of the sample indicated that they bought less products per purchase and 45.9% indicated that they bought more products per purchase in March as compared to January. A one-sample *t*-test confirmed a significant increase in purchasing quantity (*M* = 0.58, *SD* = 1.12), *t*(812) = 14.673, *p* < .001, *d* = 0.51.

Regression models were calculated for the full range scale (see S3 Table) and, for clarity of hypothesis testing regarding an increase in purchasing quantity, excluding those 8.6% of participants who report a decrease in purchasing quantity (see text below). The baseline model (see Table 2) revealed that female sex (*b* = -.19, *t*(672) = 2.05, *p* = .040, 95% CI [-.32, .02]), younger age (*b* = -.09, *t*(667) = 2.38, *p* = .018, 95% CI [-.17, -.02]) and higher education (*b* = .11, *t*(667) = 2.95, *p* = .003, 95% CI [.03, .19]) were associated with an increase in purchasing quantity. As expected, there was a positive association between *Perceived Threat of COVID-19* and change in purchasing quantity, *b* = .29, *t*(671) = 7.86, *p* < .001, 95% CI [.22, .37]. Subjects who felt more threatened by COVID-19 increased their quantity of bought products per purchase. Intolerance of uncertainty and trait-anxiety explained significant variance and were both positively associated with changes in purchasing quantity, *b* = .11, *t*(671) = 2.70, *p* = .007, 95% CI [.02, .18] respectively *b* = .10, *t*(671) = 2.42, *p* = .016, 95% CI [.01, .18]. Higher risk perception for an infection during shopping was associated with an increase in purchased quantity, *b* = .13, *t*(671) = 3.72, *p* < .001, 95% CI [.11, .27]. People who indicated to inform themselves more frequently about COVID-19 (extend of media exposure) showed in increase in purchasing quantity, b = .19, *t*(671) = 5.20, p < .001, 95% CI [.12, .26]. Belonging to a risk group was not a significant predictor of change in purchasing quantity (*b* = .03, *t*(671) = .39, *p* = .700, 95% CI [-.15, .20]) nor was having regular contact with a risk person (*b* = -.01, *t*(671) = .07, *p* = .943, 95% CI [-.16, .13]*).

Finally, all significant predictors were added to the baseline model (see Table 3). As observed for change in purchasing frequency, *Perceived Threat of COVID-19* (*b* = .22, *t*(667) = 5.02, *p* < .001, 95% CI [.13, .30]), the extend of media exposure (*b* = .12, *t(667)* = 3.10, *p* = .002, 95% CI [.05, .20]) and the perceived risk of getting infected while shopping (risk perception; *b* = .11, *t*(667) = 2.90, *p* = .004, 95% CI [.03, .20]) remained significant when adding all five variables together to the baseline model. The overall model explained 12.3% of the variance in change in purchasing quantity, *F*(10, 667) = 10.51, *p* < .001. Note that *Perceived Threat of COVID-19* (*b* = .08, *p* = .026) remained significant predictor for purchasing quantity when controlling for purchasing frequency (see S4 Table).

## Change in purchasing quantity for individual products

We analyzed the change of purchasing quantity for the individual products by entering all variables in the model. For an easier interpretation, we calculated the mean of change in purchasing quantity for "hygiene products", i.e., toilet paper, soap, and disinfectants. In the same manner ratings for pasta/rice and canned food were aggregated to form the variable "non-perishable food". For clarity of data interpretation, participants indicating that they bought less of a product were excluded from the analysis (3.3% for non-perishable food, 1.1% for hygiene products and 8.2% for fresh food). *Perceived Threat of COVID-19* (see S5 Table) was associated with an increase in purchasing quantity for non-perishable food (*b* = .21, *t*(777) = 5.56, *p* < .001, 95% CI [.14, .29]), hygiene products (*b* = .17, *t*(789) = 4.32, *p* < .001, 95% CI [.09, .25]), and fresh food (*b* = .10, *t*(737) = 2.29, *p* = .028, 95% CI [.01, .18]). Risk Perception explained additional variance only for non-perishable food (*b* = .11, *t*(777) = 2.94, *p* = .003, 95% CI [.04, .18]). High intolerance of uncertainty was associated with an increase in purchasing of non-perishable food (*b* = .10, *t*(777) = 2.77, *p* = .006, 95% CI [.03, .17]), hygiene products (*b* = .14, *t*(789) = 3.83, *p* < .001, 95% CI [.07, .21]), and fresh food (*b* = .09, *t*(737) = 2.21, *p* = .028, 95%

CI [.01, .16]). The extend of media exposure increased explained variance for non-perishable food ($b$ = .11, $t(777)$ = 3.00, $p$ = .003, 95% CI [.04, .18]) and hygiene products ($b$ = .13, $t(789)$ = 3.47, $p$ < .001, 95% CI [.05, .20]). Having regular contact to a risk person was associated with an increase in purchasing of non-perishable food ($b$ = .17, $t(777)$ = 2.33, p = .020, 95% CI [.03, .31]). Belonging to a risk group oneself also was associated positively with an increase in purchasing of non-perishable food ($r$ = .102) but did not remain significant in the multiple regression analysis. The results for the full range scale are reported as supplementary information (S6 Table).

## Perceived Threat of COVID-19

Since *Perceived Threat of COVID-19* was our main predictor of interest, we conducted an additional multiple regression analysis (see S7 Table for all predictors) on *Perceived Threat of COVID-19*. We entered the baseline model and all variables to the model that revealed a significant bivariate correlation with *Perceived Threat of COVID-19* (see S2 Table) to analyze which variables add specific variance to *Perceived Threat of COVID-19*. Female subjects indicated higher *Perceived Threat of COVID-19*, $b$ = -.33, $t(803)$ = 4.46, $p$ < .001, 95% CI [-.47, -.18]. Age showed a negative association with *Perceived Threat of COVID-19*, $b$ = -.09, t(803) = 2.66, $p$ = .008, 95% CI [-.15, -.02]. Educational level was positively related with *Perceived Threat of COVID-19*, $b$ = .13, $t(803)$ = 4.48, $p$ < .001, 95% CI [.08, .19]. Trait-anxiety ($b$ = .21, $t(803)$ = 5.44, $p$ < .001, 95% CI [.13, .29]) and risk perception ($b$ = .26, $t(803)$ = 8.30, $p$ < .001, 95% CI [.20, .32]) were positively related with *Perceived Threat of COVID-19* and added specific variance to the model. Besides, higher frequency of information gathering (media exposure) was positively associated with *Perceived Threat of COVID-19*, $b$ = .27, $t(803)$ = 8.57, $p$ < .001, 95% CI [.21, .33]. The model explained 28.3% of the variance of *Perceived Threat of COVID-19*. Note that due to the high correlation between trait-anxiety and intolerance of uncertainty ($r$ = *.61*), intolerance of uncertainty did not reach significance ($p$ = .050). Intolerance of uncertainty added incremental variance when trait-anxiety was removed from the model, $b$ = .19, $t(804)$ = 5.98, $p$ < .001, 95% CI [.13, .25].

## Discussion

The COVID-19 pandemic affected purchasing behavior all over the world. For future pandemics or a new flaring up of the COVID-19 infections it is important to understand relevant factors that influence panic buying. The aim of the study therefore was to investigate the role of *Perceived Threat of COVID-19* and anxiety related measures on purchasing behavior. So far, studies investigating the influence of threat and anxiety on changes in purchasing behavior are scarce (e.g., Garbe and colleagues who have investigated the role of threat on purchasing of toilet paper [33] and Bentall and colleagues who also used a foraging framework [34]). In the present study, we investigated the role of *Perceived Threat of COVID-19* and anxiety on purchasing behavior on a more general level and for different individual products.

The current study provides the following main findings: First and in line with our hypotheses, we found that the extend of *Perceived Threat of COVID-19* is a significant predictor for changes in purchasing behavior, i.e., high threat was associated with a tendency to buy larger quantities per purchase and a reduction in purchasing frequency in March 2020 as compared to January 2020. Second, high intolerance of uncertainty was associated with an increase in purchasing quantity but not purchasing frequency (but significant suppression effect); trait-anxiety, which was highly correlated with intolerance of uncertainty, revealed a similar pattern, although there was a significant but small correlation with purchasing frequency ($r$ = -.08). Third, participants indicating a high extend of information gathering about COVID-19 tended

to buy larger quantities and reduced purchasing frequency in March as compared to January 2020. Contrary to our expectations, being part of a risk group for a severe course of a COVID-19 infection or having contact to a person being part of such a group was not predictive for changes in purchasing behavior. All reported effects were controlled for gender, age, educational level, household size and a social desirability bias. Entering all significant predictors in one model revealed that *Perceived Threat of COVID-19* was the best predictor for change in purchasing frequency as well as for change in purchasing quantity. For change in purchasing frequency *Perceived Threat of COVID-19*, the extend of media exposure and participants' risk perception of getting infected with COVID-19 while shopping were the only predictors that remained significant. The overall analysis for change in purchasing quantity revealed the same pattern of significant effects.

The observed purchasing pattern in our study shows resemblance to the strategic behavior seen in rodents. After the experience of an electrical shock in a foraging area, animals modified their foraging behavior to reduce the possibility of experiencing an aversive event by reducing the number of entrances to the foraging area while increasing meal size [8]. According to the threat imminence model, there are three defensive modes, each associated with a specific set of behaviors [35]. The mode activated depends on predatory imminence, i.e., the probability to encounter a predator. The pre-encounter mode is the first mode in the threat continuum and is activated when entering an area indicating some predatory potential. This mode is associated with meal pattern reorganization or protective nest maintenance which can be observed in animals. Our study provides evidence that humans also show similar adaptions in the face of the threat of a virus: buying larger quantities reduces the number of visits to stores necessary to maintain food supply and thus reduces the risk of an infection in the store. Importantly, the observed pattern of purchasing behavior was also predicted by the participant´s risk perception of being infected while shopping, which was correlated positively with *Perceived Threat of COVID-19* ($r = .36$). These findings suggest that the subjective assessment of infection risk is associated with feelings of threat and influences purchasing behavior. Similar results were observed in another online survey which also used an animal foraging framework to explain changes in purchasing behavior [34]. In this study, perceived probability of getting an infection was positively associated with increased purchasing quantity. In contrast to the present study, the authors emphasized on threat due to scarcity which is not covered in our study. The moderate correlation between risk perception and *Perceived Threat of COVID-19* as found in our study suggests that additional factors–as for instance threat of scarcity–might explain additional variance in perceived threat. According to Bentall and colleagues [34], perceived risk of infection is a factor influencing scarcity vulnerability. Future studies ought to include threat due to scarcity to test whether *Perceived Threat of COVID-19* remains a meaningful predictor for changes in purchasing behavior after controlling for threat due to scarcity. Unlike the rodents in the experiments by Fanselow and colleagues [8], participants did not experience an aversive event (e.g., electrical shock). Experimental studies (instructed fear paradigms) show that next to direct experience, fear and anxiety can be acquired also by informational transmission [36, 37]. Since the outbreak of COVID-19, information about the virus and current numbers of new infections are reported on a daily basis. As reported elsewhere regular media exposure is a predictor of fear of the coronavirus [17]. In line with these findings, our analyses revealed that a greater extend of media exposure was associated with a higher level of *Perceived Threat of COVID-19* suggesting its possible role as a form of verbal instruction of threat during the corona pandemic. At the same time, media exposure was associated with an increase in purchasing quantity and a decrease in purchasing frequency. Another study, using structural equation models, revealed that cyberchondria—that is, excessive information gathering about COVID-19 combined with feelings of frustration and anxiety—is positively associated with

the intention to make unusual purchases [38]. A qualitative study on contents on twitter about toilet paper hoarding found out that nearly half of the analyzed tweets expressed negative feelings toward panic buying [39]. The authors hypothesize that this might lead to emotional distress, depression and anxiety-driven panic buying (see also [40]).

To get a more differentiated view, we also assessed the change in purchasing quantity for individual products. Our study extends the results reported by Garbe and colleagues [33] who investigated the role of perceived threat by COVID-19 and personality traits on purchasing of toilet paper. The authors found that high perceived threat by COVID-19 and high levels of emotionality predicted the amount of stockpiled toilet paper. In line with this finding, our data revealed that *Perceived Threat of COVID-19* was positively associated with an increase in purchasing quantity for non-perishable food (canned foods, pasta/rice) and hygiene products (soap, toilet paper, disinfectants). Unexpectedly, high threat was also associated with an increase in the purchasing of fresh products, although this model showed the least variance explanation (see S5 Table). Next to *Perceived Threat of COVID-19*, intolerance of uncertainty added incremental variance for all product categories indicating that anxiety as a personality trait drives changes in purchasing behavior under threat.

Although only included as control variable, we found out that female sex was associated with a decrease in shopping frequency. This result could be interpreted as a more cautious behavior in female compared to male individuals. A study investigating the role of messaging and gender on intentions to wear a face covering under COVID-19 pandemic revealed that woman more than men intend to wear a face covering [41]. A mediating factor was the subjective likelihood to get the disease, supporting our post-hoc hypothesis that women behave more cautious under the COVID-19 pandemic. Note that in our data female sex was associated with higher levels of *Perceived Threat of COVID-19*.

A limitation of this study is the retrospective rating of purchasing behavior in March 2020 which could be affected by memory biases. Longitudinal data would be important to see if subjective ratings of pre- and post-pandemic purchasing behavior differ and are associated with changes in perceived threat. The explained variance for change in purchasing frequency and purchasing quantity was rather small, indicating that additional factors were associated with a change in purchasing behavior. Recent studies indicate that e.g., right political affiliation [13, 34], the extend of engaging in social distancing [13], and higher levels of paranoia [34] are associated with more stockpiling. Due to the correlational nature of this study no claims about causality can be made. Therefore, we cannot rule out that the found correlations between purchasing behavior and *Perceived Threat of COVID-19* are coincidental although data from the German Federal Statistical Office suggests that there was indeed an unusual increase in sales figures in March compared to the mean of August 2019 to January 2020 [3]. More experimental studies should try to translate findings from animal experiments to human behavior to test whether certain behaviors are associated with different threat levels as reported in a study by Bach and colleagues [42]. Such studies could provide further evidence that foraging behavior is a relevant behavioral component of anxiety and fear in humans. Another limitation is the sex bias observed in the current study (78% of respondents were female) and the high proportion of high educated respondents which reduces generalizability although we controlled for sex and educational level. Two major strengths of this study can be mentioned: First, the derivation of hypotheses based on an animal model contributes to link findings from animal literature to human behavior. Second, this study collected purchasing behavior, anxiety ratings and *Perceived Threat of COVID-19* around the peak of the COVID-19 pandemic in Germany and thus provides unique data about behavior and related predictors under an extreme event.

## Conclusion

In conclusion perceived *Perceived Threat of COVID-19* influences purchasing behavior in a twofold way: high levels of threat are associated with an increase in purchasing quantity and a reduction in purchasing frequency. The positive relation between *Perceived Threat of COVID-19* and an increase of purchasing quantity was confirmed for individual products, too. Next to the *Perceived Threat of COVID-19*, intolerance of uncertainty and the level of perceived risk for an infection during shopping also were significant predictors for purchasing behavior (quantity and frequency). While intolerance of uncertainty might be a relative stable personality trait, a reduction of risk perception could help to mitigate maladaptive changes in purchasing behavior like panic buying. Our data suggests that the extend of media exposure is associated with feeling of threat and change in purchasing pattern. This highlights the importance of appropriate risk communication. Information about effective protection measures while shopping could reduce high risk perception of being infected during shopping and might help to prevent panic buying. Additionally, recommendations about the amount of information gathering in media could have beneficial effects (e.g., informing only once per day to reduce negative effects).

## Supporting information

**S1 Appendix. German version of the online questionnaire.**
(DOCX)

**S2 Appendix. English version of the online questionnaire.**
(DOCX)

**S3 Appendix. Non-parametric data analysis.**
(DOCX)

**S1 Table. Bivariate correlations.**
(DOCX)

**S2 Table. Bivariate correlations for the full range scale.**
(DOCX)

**S3 Table. Multiple regression analysis for the full range scale.**
(DOCX)

**S4 Table. Multiple regression analysis controlling for change in purchasing frequency.**
(DOCX)

**S5 Table. Multiple regression analysis for individual products.**
(DOCX)

**S6 Table. Multiple regression analysis for individual products for the full range scale.**
(DOCX)

**S7 Table. Multiple regression analysis for Perceived Threat of COVID-19.**
(DOCX)

## Author Contributions

**Conceptualization:** Sebastian Schmidt, Christoph Benke, Christiane A. Pané-Farré.

**Formal analysis:** Sebastian Schmidt.

**Methodology:** Sebastian Schmidt, Christoph Benke, Christiane A. Pané-Farré.

**Writing – original draft:** Sebastian Schmidt.

**Writing – review & editing:** Sebastian Schmidt, Christoph Benke, Christiane A. Pané-Farré.

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
