## [Decision Letter · Decision Letter 0]

10 Mar 2021

PONE-D-21-01441

Purchasing under threat: Changes in purchasing patterns during the COVID-19 pandemic

PLOS ONE

Dear Dr. Schmidt,

Thank you for submitting your manuscript to PLOS ONE. After careful consideration, we feel that it has merit but does not fully meet PLOS ONE’s publication criteria as it currently stands. Therefore, we invite you to submit a revised version of the manuscript that addresses the points raised during the review process.

Three experts in the field read your manuscript and returned a positive feedback. I agree with them that your translational approach for this relevant topic is interesting and inspiring. However, they also erased several questions regarding the analysis and the discussion of the results. I am therefore asking you to consider their valuable comments as well as the numerous suggestions about recent literature.

We look forward to receiving your revised manuscript.

Kind regards,

Marta Andreatta

Academic Editor

PLOS ONE

Journal Requirements:

2. Please include additional information regarding the survey or questionnaire used in the study and ensure that you have provided sufficient details that others could replicate the analyses.

For instance, if you developed a questionnaire as part of this study and it is not under a copyright more restrictive than CC-BY, please include a copy, in both the original language and English, as Supporting Information.

3. Please note that PLOS ONE does not copy edit accepted manuscripts (https://journals.plos.org/plosone/s/criteria-for-publication#loc-5).

To that effect, please ensure that your submission is free of typos and grammatical errors.

Reviewers' comments:

Reviewer's Responses to Questions

**Comments to the Author**

1. Is the manuscript technically sound, and do the data support the conclusions?

Reviewer #1: Yes

Reviewer #2: Yes

Reviewer #3: Yes

2. Has the statistical analysis been performed appropriately and rigorously? 

Reviewer #1: Yes

Reviewer #2: Yes

Reviewer #3: Yes

3. Have the authors made all data underlying the findings in their manuscript fully available?

Reviewer #1: Yes

Reviewer #2: Yes

Reviewer #3: Yes

4. Is the manuscript presented in an intelligible fashion and written in standard English?

Reviewer #1: Yes

Reviewer #2: Yes

Reviewer #3: Yes

5. Review Comments to the Author

Reviewer #1: Review: Purchasing under threat: Changes in purchasing patterns during the COVID-19

pandemic

In this study, the authors investigated changes in purchasing patterns during the first wave of the COVID-19 pandemic in Germany in a retrospective online survey. Results show that high levels of perceived COVID-19-threat, intolerance of uncertainty and media exposure were associated with reduced purchase frequency and increased number of products bought. Moreover, results have been linked to animal models of foraging behavior under threat.

The study background, procedure, results and discussion are concise and clear; data are freely available. The results of the present survey are of broad interest. I do have only some minor suggestions.

Title:

I would recommend a more eye-catching title avoiding repetition of “purchasing” (e.g., Shopping or buying)

Introduction:

Please cite international data for “worldwide panic buying” (p. 3, line 1).

Methods:

p. 4: May_18th (missing space)

Please provide more detailed information on the online assessment method used (software; e.g. EvaSys; SociSurvey).

I would stick to the previous wording “Perceived Threat of COVID-19 in the whole manuscript (p. 6, Predictors […])

p. 7, line 14f: description of the Likert scale has no quotation marks (unlike the previous scales), see also Demographic Variables below (I would stick to one formatting)

Results:

p.16, line 9 and p. 14, lines 23 and 25 and p. 15 lines 1 and8: p <_.001 (missing spaces)

Discussion:

I would recommend including further research in a more detailed discussion. For example, Leung et al. (2021) recently published an interesting analysis on social media content, highlighting the role of social media in anxiety and panic buying behaviors (see p.17, line 4f). Also, there is a publication by Bentall et al. (2021) on pandemic buying which is also based on animal foraging theory, which should be included. (see also Lehberger et al., 2021; Micalizzi et al, 2020).

When targeting the “Threat of COVID-19”, it would be interesting whether reducing the number (or encounter probability) of “predators” (which is, other potentially contagious individuals in the human model) in the supermarket might reduce perceived threat. Is there any animal research paralleling this restriction measure (already taken by several countries)?

Reviewer #2: To be honest, I found this paper very well written. Easy to read and to grasp. I do not have any major comment. I just took a couple of minor notes, while reading it:

- The “perspective article” on what social and behavioural science can do to support pandemic response, publishes by Van Bavel et al. in Nature Human Behaviour can be a useful general reference for the Introduction.

- “As observed in animal studies, increased threat, i.e., increased probability of the predator entering the foraging area (as indicated by a visual cue) led to a reduction of the number of tokens being collected while participants stayed closer to the safe space, to increase chances of successful flight from predatory attack [10]”. My understanding is that there is something missing here, that is, the fact that the amount of money collected each time is greater. Perhaps they did not find this, simply because participants do not need money to actually survive. So, it’s possible that this setting is quite different from the setting under study in the paper.

- The result on gender differences in purchasing behaviour under threat could be related to the emerging literature on gender differences in COVID-19 preventative behaviours (Capraro & Barcelo, 2020; Coroiu et al. 2020; Haischer et al. 2020).

References

Capraro, V., & Barcelo, H. (2020). The effect of messaging and gender on intentions to wear a face covering to slow down COVID-19 transmission. Journal of Behavioral Economics for Policy, 4, Special Issue 2, 45-55.

Coroiu, A., Moran, C., Campbell, T., & Geller, A. C. (2020). Barriers and facilitators of adherence to social distancing recommendations during COVID-19 among a large international sample of adults. PloS one, 15(10), e0239795.

Haischer, M. H., Beilfuss, R., Hart, M. R., Opielinski, L., Wrucke, D., Zirgaitis, G., ... & Hunter, S. K. (2020). Who is wearing a mask? Gender-, age-, and location-related differences during the COVID-19 pandemic. PloS one, 15(10), e0240785.

Van Bavel, J. J., et al. (2020). Using social and behavioural science to support COVID-19 pandemic response. Nature Human Behaviour, 4, 460-471.

Reviewer #3: Thank you for the opportunity to read and comment on this manuscript.

In this online study, the associations between perceived threat and risk of COVID-19, media exposure, further variables (e.g., intolerance of uncertainty, trait anxiety) and purchasing behavior are investigated. The authors try to explain and compare their results with foreaging behavior observed in animals. As stated by the authors, studying factors which influence behavioral responses to the COVID-19 pandemic is of high relevance. The present study contributes to this increasing body of research by focusing on the investigation of purchasing behavior. Major strengths of the study are the large sample size (possibly due to the online-format of the study), and the inclusion and parallel investigation of different variables all possibly associated with purchasing behavior.

Nevertheless, I have the following questions and suggestions that might be addressed in a possible revision of this manuscript:

1. Although I welcome a concise introduction, I still think that it would benefit from a deeper and more precise theoretical deduction of hypotheses on the association between anxiety psychopathology (intolerance of uncertainty, trait anxiety), media exposure and threat of COVID-19 or behavioral responses (e.g., for intolerance of uncertainty: Tull et al., 2020 [10.1016/j.janxdis.2020.102258]; Gertens et al., 2020 [10.1016/j.janxdis.2020.102258]). A short presentation of results of previous research on purchasing behavior and associated factors during the COVID-19 pandemic (e.g., Micalizzi et al., 2020 [10.1111/bjhp.12480] or quite new results from Bentall et al., 2021 [10.1371/journal.pone.0246339]), which also used a animal foreaging approach) would be likewise interesting. Furthermore, I would like to see the authors name all hypotheses (hypotheses on the explanation of threat of COVID-19; risk for oneself/loved ones) and especially the direction of all studied hypotheses (e.g., expected direction of associations of intolerance of uncertainty/trait anxiety and purchasing behavior). This, then, also refers to the statement in the discussion, that some results have been unexpected (e.g., no association between being part of a risk group and purchasing behavior).

2. Additional analyses / hypotheses:

a. Explaining human purchasing behavior in the context of a pandemic situation with animal foreaging models is an interesting approach. The cited study on human behavior under threat (Bach et al., 2014) (introduction) showed – in my opinion – only limited comparability to the cited animal studies, as the predator in this case did not endanger the “life” of people but “only” the collected rewards, which possibly overlaps more with an assumed shortage of goods in a pandemic situation. It, therefore, made me think, if the change of purchasing behavior in a pandemic situation could possibly be better or additionally explained by a presumed risk of shortage of goods than by the perceived threat of COVID-19 while buying. The authors do refer to this alternative hypothesis in the discussion section. I was still wondering whether the authors assessed and controlled for this aspect in the present study?

b. Purchasing frequency and quantity are highly correlated in the present study. This absolutely makes sense, as one would assume that, if going less often grocery shopping, you would have to buy more when going grocery shoping. Did the authors perform regression analyses on purchasing quantity, which controlled for purchasing frequency?

c. Explained variance of purchasing frequency and quantity could be considered as rather small in all analyses. Reflecting on that, I asked myself, which other variables could contribute to purchasing behavior in a pandemic situation. Additionally to the assumption already mentioned in a.), I still wondered, if their could also have been effects of the amount of staying-at-home-time on purchasing quantity (e.g., no canteen during lunch break, no visits of restaurants). Did the authors ask the participants to rate the amount of home-prepared meals? If not so, I still would appreciate a short reflection on the “rather low” explained variance and other possibly influencing factors in the discussion.

d. The authors excluded people, who, e.g., have been infected with COVID-19 or who did not buy their purchases before the COVID-19 pandemic, which can be considered as plausible. I am still interested if the authors did analyze, even on an exploratory purpose, the associations between perceived threat of COVID-19 and the decision to not make one’s own purchases (n = 150) or to exclusevily buy online (n = 62). One could assume that especially the latter group would show higher levels of perceived threat of COVID-19.

e. The separate calculation of regression models in the total sample and in groups reporting a decrease or stability of purchase frequency and in groups reporting an increase or stability of purchase quantity seems to be plausible considering the proposed hypotheses. I highly welcome the additional reporting of results of the full range scale as a supplementary table. I, nevertheless, asked myself, if the authors also have calculated the “total regression model” (likewise to Table 3) of the full range scale of purchasing frequency and quantity and regression models on the full range scale of purchasing quantity for individual products. Did the reported results and conclusions stay stable?

3. Data analysis:

a. Model assumptions / Robustness of results:

i. I was wondering if the authors did check for regression model assumptions? Detected medium correlations between some predictor variables (S1 Table 1, Table 2) could, e.g., indicate some multicollinearity of predictor variables.

ii. Nevertheless, in order to gain more robust results, did the authors consider using bootstrapping? If not so, I would still suggest adding confidence intervals and standard errors for all regression estimates.

b. Data level:

i. As the outcomes “purchasing frequency” and “purchasing quantity” rather are ordinal variables, I was wondering if the authors considered using non-parametric tests for calculation of differences of central tendencies (e.g., Wilcoxon) or for performing regression analyses (ordinal regression)? Regarding the paired samples t-tests, I would highly recommend to compute and report effect size measures (e.g., Cohen’s d)

ii. Correlational analyses: Again, considering the different data structure of the reported variables, I was asking myself if the authors took this into account when computing correlational analyses (e.g., Spearman Rho)?

c. The authors have not commented on the handling of missing data yet. Has their been missing data and how have the authors proceded with missing data?

4. Minor comments:

a. Abstract: I would propose to add exact statistics (p-values, b) for the main reported effects (associations between threat of COVID-19 and purchasing frequency and quantity, associations between intolerance of uncertainty and extend of media exposure and threat of COVID-19)

b. Paragraph 2 on page 3: The sentence “Based on a theoretical framework […]” sounds more like a summary of an aim or like a introduction to the presentation of hypotheses. Therefore, I would suggest to move this statement and either include it as an introduction to your hypotheses or add it to the discussion.

c. In the first paragraph of the method section (pages 4-5), the authors refer to the implementation of public health measures, and to an increase of risk communication and sales figures in Germany. I would consider it helpful to include references and examples for these statements.

d. As perceived threat of COVID-19 serves as the main predictor in the present study, I would like to see the authors name all six items which constitute the mean score. I might have missed it, but did the authors name the scaling of the rating scales of perceived threat of COVID-19 in the measures section?

e. As I assume that the authors used the German versions of the presented measurement instruments, I would suggest adding the respective references and reporting the internal consistencies from “German” validation studies. Additionally, I am missing a comment on validity of used instruments in previous research.

f. To increase the reading flow, please consider reviewing the manuscript on consistent naming, spelling, and highlighting (e.g., italics) of central concepts (e.g., trait-anxiety vs. trait anxiety, behaviour vs. behavior, COVID-19-Threat vs. threat of COVID-19, sex vs. gender)

g. I would like the authors to review all tables on consistent and “correct” adding of zeros before the decimal point (b and p-values)

h. Table 2 and 3 report results on both purchasing frequency and quantity. As stated by the authors before, they have used a restricted sample for the analyses reported in table 2 and 3. In the footnotes of the tables, N is, nevertheless, stated as being the same. I would like the authors to review the footnotes of the tables 2 and 3.

i. Although the coding of the educational level is explained in the methods section, an additional footnote on the coding of educational level in Table 1 would be beneficial.

6. PLOS authors have the option to publish the peer review history of their article (what does this mean?). If published, this will include your full peer review and any attached files.

Reviewer #1: No

Reviewer #2: No

Reviewer #3: No

---

## [Author Response · Author response to Decision Letter 0]

24 Apr 2021

We would again like to thank the reviewers for their thoughtful comments and suggestions. Please check the uploaded file "Response to Reviewers" for detailed answers. We hope that we have addressed all the remaining points and we also hope that you and the reviewers will find the manuscript suitable for publication in PLOS ONE. Thank you for giving this paper your consideration.

---

## [Decision Letter · Decision Letter 1]

1 Jun 2021

Purchasing under threat: Changes in shopping patterns during the COVID-19 pandemic

PONE-D-21-01441R1

Dear Dr. Schmidt,

We’re pleased to inform you that your manuscript has been judged scientifically suitable for publication and will be formally accepted for publication once it meets all outstanding technical requirements.

Kind regards,

Marta Andreatta

Academic Editor

PLOS ONE

Additional Editor Comments (optional):

I have re-read this very interesting manuscript as well as all the comments and responses.

I found that the authors properly addressed all the issues erased by the reviewers providing additional insights about the results.

I am very happy to accept this manuscript for publication.

Reviewers' comments:

Reviewer's Responses to Questions

**Comments to the Author**

1. If the authors have adequately addressed your comments raised in a previous round of review and you feel that this manuscript is now acceptable for publication, you may indicate that here to bypass the “Comments to the Author” section, enter your conflict of interest statement in the “Confidential to Editor” section, and submit your "Accept" recommendation.

Reviewer #1: All comments have been addressed

Reviewer #2: All comments have been addressed

Reviewer #3: All comments have been addressed

2. Is the manuscript technically sound, and do the data support the conclusions?

Reviewer #1: Yes

Reviewer #2: Yes

Reviewer #3: Yes

3. Has the statistical analysis been performed appropriately and rigorously? 

Reviewer #1: Yes

Reviewer #2: Yes

Reviewer #3: Yes

4. Have the authors made all data underlying the findings in their manuscript fully available?

Reviewer #1: Yes

Reviewer #2: Yes

Reviewer #3: Yes

5. Is the manuscript presented in an intelligible fashion and written in standard English?

Reviewer #1: Yes

Reviewer #2: Yes

Reviewer #3: Yes

6. Review Comments to the Author

Reviewer #1: The authors have addressed all previous points and I now recommend this very interesting, clear and concise manuscript for publication in PLOSone.

Reviewer #2: The authors have addressed all my comments, I think that the paper is now ready for publication.

Reviewer #3: Thank you for this very profound and detailed revision of the previously submitted, and yet very good manuscript. I really value that the authors have answered all my questions and have done some additional, in-depth analyses, which - I know - resulted in some workload. I think that this manuscript is very well-written and provides interesting and profound results on the behavioral response (that is purchasing behavior) to the COVID-19 pandemic. I, therefore, highly recommend to accept the manuscript. Thank you again for your answers to my comments!

7. PLOS authors have the option to publish the peer review history of their article (what does this mean?). If published, this will include your full peer review and any attached files.

Reviewer #1: No

Reviewer #2: No

Reviewer #3: No

---

## [Editor Report · Acceptance letter]

4 Jun 2021

PONE-D-21-01441R1 

Purchasing under threat: Changes in shopping patterns during the COVID-19 pandemic 

Dear Dr. Schmidt:

I'm pleased to inform you that your manuscript has been deemed suitable for publication in PLOS ONE. Congratulations! Your manuscript is now with our production department. 

Kind regards, 

on behalf of

Dr. Marta Andreatta 

Academic Editor

PLOS ONE